# Ultra-high-field imaging reveals increased whole brain connectivity underpins cognitive strategies that attenuate pain

Enrico Schulz[1,2]*, Anne Stankewitz[2], Anderson M Winkler[1,3], Stephanie Irving[2], Viktor Witkovský[4], Irene Tracey[1]

[1]Wellcome Centre for Integrative Neuroimaging, FMRIB, Nuffield Department of Clinical Neurosciences, University of Oxford, Oxford, United Kingdom; [2]Department of Neurology, Ludwig-Maximilians-Universität München, Munich, Germany; [3]Emotion and Development Branch, National Institute of Mental Health, National Institutes of Health, Bethesda, United States; [4]Department of Theoretical Methods, Institute of Measurement Science, Slovak Academy of Sciences, Bratislava, Slovakia

**Abstract** We investigated how the attenuation of pain with cognitive interventions affects brain connectivity using neuroimaging and a whole brain novel analysis approach. While receiving tonic cold pain, 20 healthy participants performed three different pain attenuation strategies during simultaneous collection of functional imaging data at seven tesla. Participants were asked to rate their pain after each trial. We related the trial-by-trial variability of the attenuation performance to the trial-by-trial functional connectivity strength change of brain data. Across all conditions, we found that a higher performance of pain attenuation was predominantly associated with higher functional connectivity. Of note, we observed an association between low pain and high connectivity for regions that belong to brain regions long associated with pain processing, the insular and cingulate cortices. For one of the cognitive strategies (safe place), the performance of pain attenuation was explained by diffusion tensor imaging metrics of increased white matter integrity.

*For correspondence:
es@pain.sc

Competing interests: The authors declare that no competing interests exist.

## Introduction

An increased perception of pain is generally associated with increased cortical activity; this has been demonstrated in a number of brain regions and processes involved in sensory, emotional, cognitive, and affective aspects of pain (*Coghill and Eisenach, 2003*; *Tracey and Mantyh, 2007*).

Given the threatening nature of pain, the information processed from these different aspects has to be integrated and assessed to compute an appropriate decision and subsequent action (*Liang et al., 2013*; *Misra and Coombes, 2015*; *Wiech and Tracey, 2013*). To achieve this goal, brain regions are required to exchange information; when pain is worsened by emotional or attentional shifts, this has been shown to entail increased functional connectivity between relevant cortical and subcortical regions (*Sprenger et al., 2015*; *Villemure and Bushnell, 2009*). Less is known, however, about regional brain connectivity changes during decreased pain experiences, despite some early efforts (*Ploner et al., 2011*).

Several studies have investigated the influence of pain on Independent Component Analysis (ICA)-based functional networks (*Kucyi et al., 2013*; *Seminowicz and Davis, 2007*). However, a different approach is required if we are to investigate the full extent of network connectivity changes. To date, this has been attempted by quantifying the covariation of the seed-based fluctuating blood-oxygen-level dependent (BOLD) activity between a priori chosen brain regions. Changes in this covariation of cortical signals have been linked to conditions that represent different levels of

pain experience. *Villemure and Bushnell, 2009* and *Ploner et al., 2011*, for example, investigated the influence of different levels of emotion and attention, respectively, on cortical connectivity between brain regions involved in pain and emotional processing. Both studies observed an increase of connectivity for the conditions that increased the intensity of pain.

A different study found that a change in pre-stimulus cortical connectivity patterns from the anterior insula to the periaqueductal grey (PAG), which is part of the descending pain modulatory system (*Sprenger et al., 2018*), determined whether a subsequent nociceptive stimulus was perceived as painful or not (*Ploner et al., 2010*). Supporting that observation, other investigations have similarly reported an increased activity or functional connectivity between the PAG and the rostral anterior cingulate cortex (rACC) for conditions associated with decreased pain intensity perception (e.g. placebo, shift of attention) (*Bantick et al., 2002*; *Eippert et al., 2009*; *Sprenger et al., 2012*; *Tracey et al., 2002*). These functional connections are likely supported by the structural integrity of white matter tracts between brain regions, as measured using diffusion tensor imaging (DTI). Indeed, a study has confirmed that the structural integrity of components of the descending pain modulatory system were significantly correlated to the effectiveness in alleviating pain through transcranial direct current stimulation (*Lin et al., 2017*).

A number of studies point to the relevance of connectivity changes in pain modulation – largely, but not exclusively, centred on regions within the descending pain modulatory system. However, the precise nature of connectivity changes within the whole brain, especially during decreased pain, is still unclear. Using ultra-high-field functional magnetic resonance imaging (fMRI) that facilitates single-trial analysis, we examined the whole-brain functional connections that contribute to the attenuation of pain. We used three different cognitive interventions: (a) a non-imaginal distraction by counting backwards in steps of seven; (b) an imaginal distraction by imagining a safe place; and (c) reinterpretation of the pain valence (cognitive reappraisal). These cognitive strategies are hypothesised to be represented in the brain by a complex cerebral network that connects a number of brain regions, where:

1. The effective use of a cognitive strategy that is successful for pain attenuation results in increased functional connectivity between task-related brain regions.
2. Decreased connectivity is expected between cortical areas that are involved in the processing and encoding of pain intensity, for example sub-regions of the insular cortex, the cingulate cortex, somatosensory cortices, and PAG.
3. Increased connectivity is hypothesised for the descending pain control system, particularly for the connection between the rACC and the PAG.
4. Altered connectivity from insula sub-divisions to frontal and somatosensory regions will result as a consequence of their high relevance in integrating sensory information.

Healthy participants were asked to utilise cognitive strategies in order to attenuate the experience of pain during 40 s of cold stimulation. We pursued a whole-brain parcellation approach (*Glasser et al., 2016*) in order to assess every cortical connection that contributes to pain relief.

## Results

Behavioural data revealed a significant reduction in the ratings of perceived pain for all three interventions compared to the unmodulated pain condition. By using a numerical and a visual analogue scale (VAS), ranged between 0 and 100, the participants rated the stimuli as significantly less painful compared to the unmodulated pain condition. For example, for the counting condition we observed a pain attenuation of 21 (±2.55 SE) for pain intensity and of 19 (±2.87 SE) for pain unpleasantness ($p < 0.05$). More detailed results are reported in our previous publication (*Schulz et al., 2019*).

Globally, we found increased connectivity during pain attenuation: pain during the cognitive task trials was rated as less painful and had a stronger global brain connectivity compared to trials where there was no cognitive modulation (i.e. the unmodulated pain condition). Therefore, trials at higher levels of pain are coupled with low connectivity, and trials at lower levels of pain are coupled with high connectivity.

We pursued a whole-brain approach by subdividing the cortex into 180 regions per hemisphere, plus 11 subcortical regions (*Glasser et al., 2016*), and related cortical connectivity to pain ratings at a single trial level. This approach was facilitated by the increased signal-to-noise as a result of ultra-high field imaging, as well as by a more robust assessment of single-trial data from longer lasting

painful stimulation and an extended task application. For analysis of the three conditions, we merged the cognitive intervention trials with the unmodulated pain trials. This has two major advantages:

i. First, it takes the within-subjects variable performance of the pain attenuation attempts into account; for example a more effective attempt to attenuate pain is considered to cause a different cortical connectivity than a less effective attempt.
ii. Second, we also take into account the more natural fluctuation of the unmodulated pain trials.

The findings are represented in confusion matrices, depicting the pain intensity-related connectivity between all brain regions. Some of the regions were previously interpreted as task-related and others as belonging to pain processing regions and circuits (see *Schulz et al., 2019*). In the present study, positive relationships (*red*) show connectivities that were increased in particularly effective pain attenuation trials (performance encoding). Across all tasks, we confirmed our first hypothesis by showing that increased connectivity of *task-processing* brain regions (see *Schulz et al., 2019*) is related to particularly effective attempts to attenuate pain. Negative relationships (*blue*) represent cortical connections that are disrupted in effective pain attenuation trials. Disruptions were hypothesised to occur for the *core regions of pain processing*, such as for the various subregions of the insular, cingulate, and somatosensory cortices. However, we found that these regions predominantly showed increased connectivity during effective pain attenuation trials. Further, we investigated the connectivity of those brain regions belonging to the neurological signature of physical pain processing (NPS; *Wager et al., 2013*). For this additional analysis, we averaged 40 bilateral insular, opercular, and cingulate regions, the bilateral thalamus plus the PAG. We averaged the connectivity across these 81 "NPS" regions for each single trial across modulated and unmodulated trials.

## Counting

We investigated changes in connectivity patterns related to effective trials during counting. Out of 68635 potential functional connections, we found 171 connections were significantly increased or decreased during the counting condition. Although it is not possible to readily illustrate all these significant connections, a prominent pattern is visible: we found a general increase of cortical connectivity with the exception of decreased connections involving the right temporo-parieto-occipital junction (*Figure 1A*). Some regions show a particularly strong connectivity: the right anterior and posterior insula, the left and right temporal cortices, the left parietal cortex, as well as higher order visual regions in occipito-temporal areas. The most connected area is the right middle insula (R_MI, *Figure 1C*; for the nomenclature of these regions see *Glasser et al., 2016*). Our earlier publication reported that the right middle insula was decreased in activity during pain attenuation (see *Schulz et al., 2019*). We suggest it might suppress activity in the posterior cingulate cortex (L_23 c, R_23 c). There are prominent connectivity patterns, particularly from parietal regions, that could result in the suppression of right insula and subsequently posterior cingulate activity. There are intrahemispheric connectivity increases between subdivisions of the insular cortices but, interestingly, no increases between subdivisions of the right insula itself. In addition, areas in the left medial wall of the parietal cortex (Brodmann area 7) are functionally connected to a right posterior cortical region that stretches from higher order visual areas (lateral occipital cortex) to the posterior medial temporal cortex. The homologue left occipito-temporal region is functionally connected to the right inferior parietal lobe (subregions PFt and PFop). Regions in the left superior and middle temporal cortex are strongly connected with several sections of the insular cortex (temporal regions STS and TE1). Extended regions in the left superior parietal cortex (Brodmann area 5) and the posterior cingulate cortex are functionally connected with the right middle insular cortex (*Figure 1C and D*). There was no significant change in connectivity to subcortical areas. The detailed matrix of statistical results can be found in the source data files (*Source data 1*). We confirmed an increased functional connectivity with pain attenuation for the counting condition in the network of areas related to the "NPS" (p<0.05, uncorrected).

Interestingly, measures of structural connectivity (DTI fibre tracking) did not explain interindividual differences in modulating task-related functional connections in the counting condition.

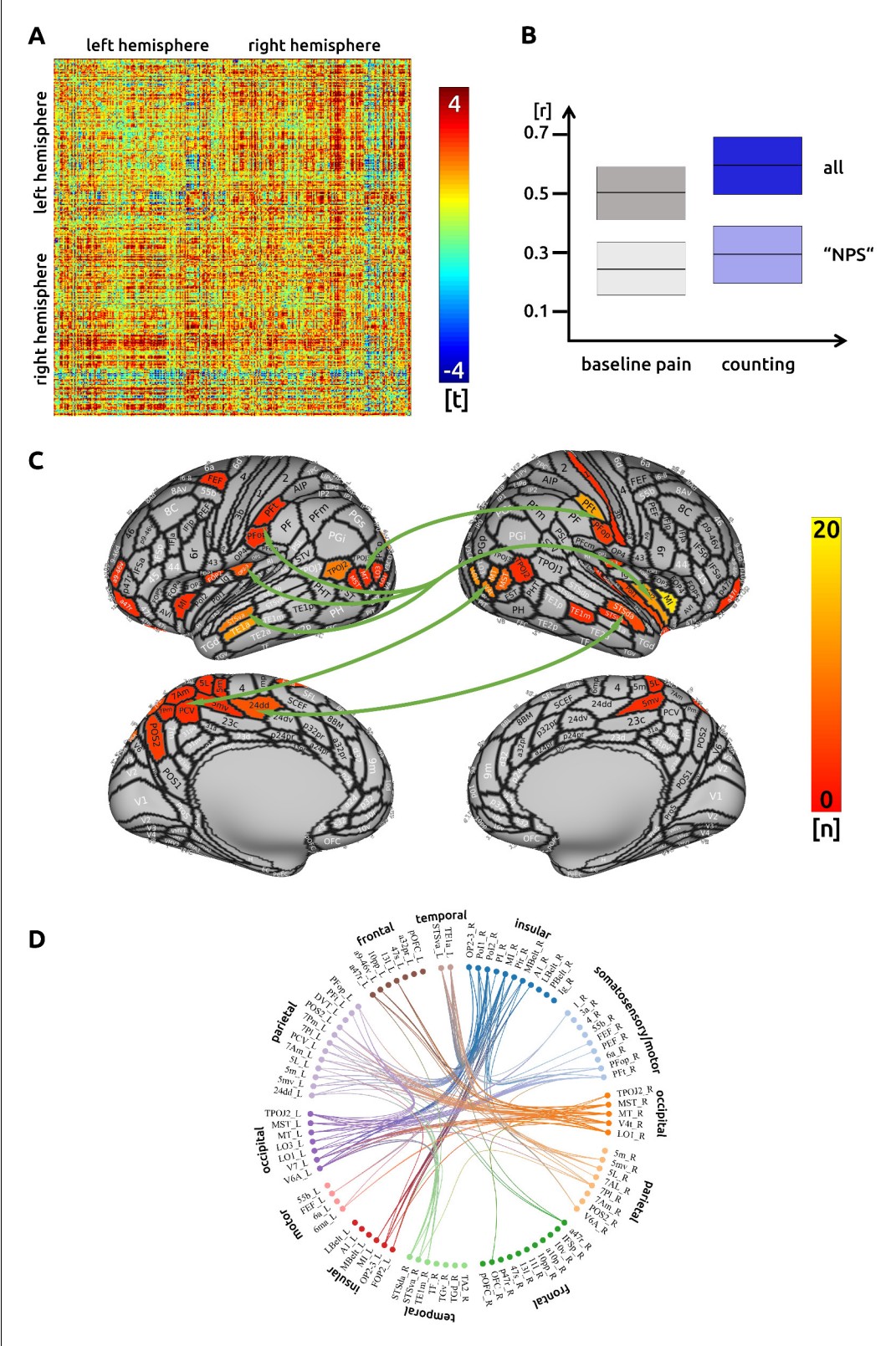

**Figure 1.** Counting: (**A**) the confusion matrix shows the statistical results. Each line represents one of the 371 ROIs. The values are mirrored along the principal diagonal of the matrix. A single red dot represents the varying connectivity between two specific brain regions and indicates that a stronger cortical connectivity in a single trial is related to a decrease in pain perception (performance encoding). These findings are the result of the higher connectivity in the trials of the counting task compared to unmodulated pain trials. (**B**) Data from the confusion matrix averaged across all subjects,

*Figure 1 continued on next page*

*Figure 1 continued*

connections and trials (mean ± standard deviation; for illustration purposes only). The darker boxes show the average across all connections, the lighter boxes represent the averaged connections within the "NPS". (C) Depiction of the cortical regions as defined by the Glasser parcellation; the arrows show the best-connected regions; the right middle insular cortex has the most connections where connectivity changes are shown to significantly modulate pain intensity. Only regions with at least three significant connections (n > 2) are included in the cortical map. (D) For the circular plot, created with Brain Data Viewer (link) we selected 89 regions that showed at least three significant connections in any of the three conditions. For more detailed information on the exact connections see the source data files.

## Safe place

We found 210 functional connections that were significantly increased or decreased during the imagining condition. We found an increase of connectivity across all cortical regions when compared to the unmodulated pain condition (*Figure 2A*). There was no negative relationship between single-trial connectivity and pain intensity.

For this safe condition, we again found that a main hub for functional connectivity is the right insular cortex. Besides this region, we observed attenuation-related connectivity changes in right parietal (BA 5) and left superior parietal cortices (BA 7). Further well-connected areas include a frontal language area (BA 55b), as well as motor and premotor areas. Regions in the right posterior insular cortex are connected to the left parietal cortex (BA 7). The right precentral areas are functionally interconnected with prefrontal and orbitofrontal areas, the right parietal cortex (BA 5), and the left superior parietal cortex (BA 7). The right 'belt' regions are functionally connected to prefrontal and orbitofrontal areas (*Figure 2C and D*). There was no significant connectivity change to subcortical areas. The detailed matrix of statistical results can be found in the source data files (*Source data 2*). We confirmed an increased functional connectivity with pain attenuation for the 'safe place' condition in the network of areas related to the "NPS" (p<0.001, uncorrected).

For this 'safe place' condition only, we found that white matter structural connectivity, as measured using DTI, mediates the strength of the functional connectivity (*Source data 4_DTI*). Strong structural connectivities are related to a better ability to modulate the functional connectivity in order to attenuate pain. This applies especially to connections between frontal regions (IFSp and Brodmann area 8C) and the secondary somatosensory cortex (SII). Further functional connections that are supported by the strength of fibre connections, projected to memory-related areas (presubiculum of the hippocampus and entorhinal cortex).

## Reappraisal

While executing cognitive reappraisal, we found a pain attenuation-related increase of functional connectivity compared to the unmodulated pain condition across the entire cortex (*Figure 3A*). Decreased functional connectivity was not observed. Connections that included frontal premotor and insular sub-regions contributed to a decrease of pain (*Figure 3C and D*). However, the main hub of connectivity was located in the medial parieto-occipital cortex. Besides other regions, the area V6A is interconnected with several insular and frontal premotor areas, some of which control eye movements. The structural characteristics between cortical regions did not contribute to an enhanced functional connectivity for reappraisal. Again, there was no significant connectivity change to subcortical areas. The detailed matrix of statistical results can be found in the source data files (*Source data 3*). We confirmed an increased functional connectivity with pain attenuation for the reappraisal condition in the network of areas related to the "NPS" (p<0.005, uncorrected).

## Conjunction analysis

We did not find any connectivity changes related to pain modulation that were present in all three conditions. The direct comparison of the three conditions exhibited functional connections that were singularly present for each condition (see source data files 5 to 7). For the counting condition, we found 63 regions that had a stronger effect compared to the other two conditions. Similarly, 7 regions for 'safe place' and 11 regions for reappraisal were more strongly connected compared to the other conditions.

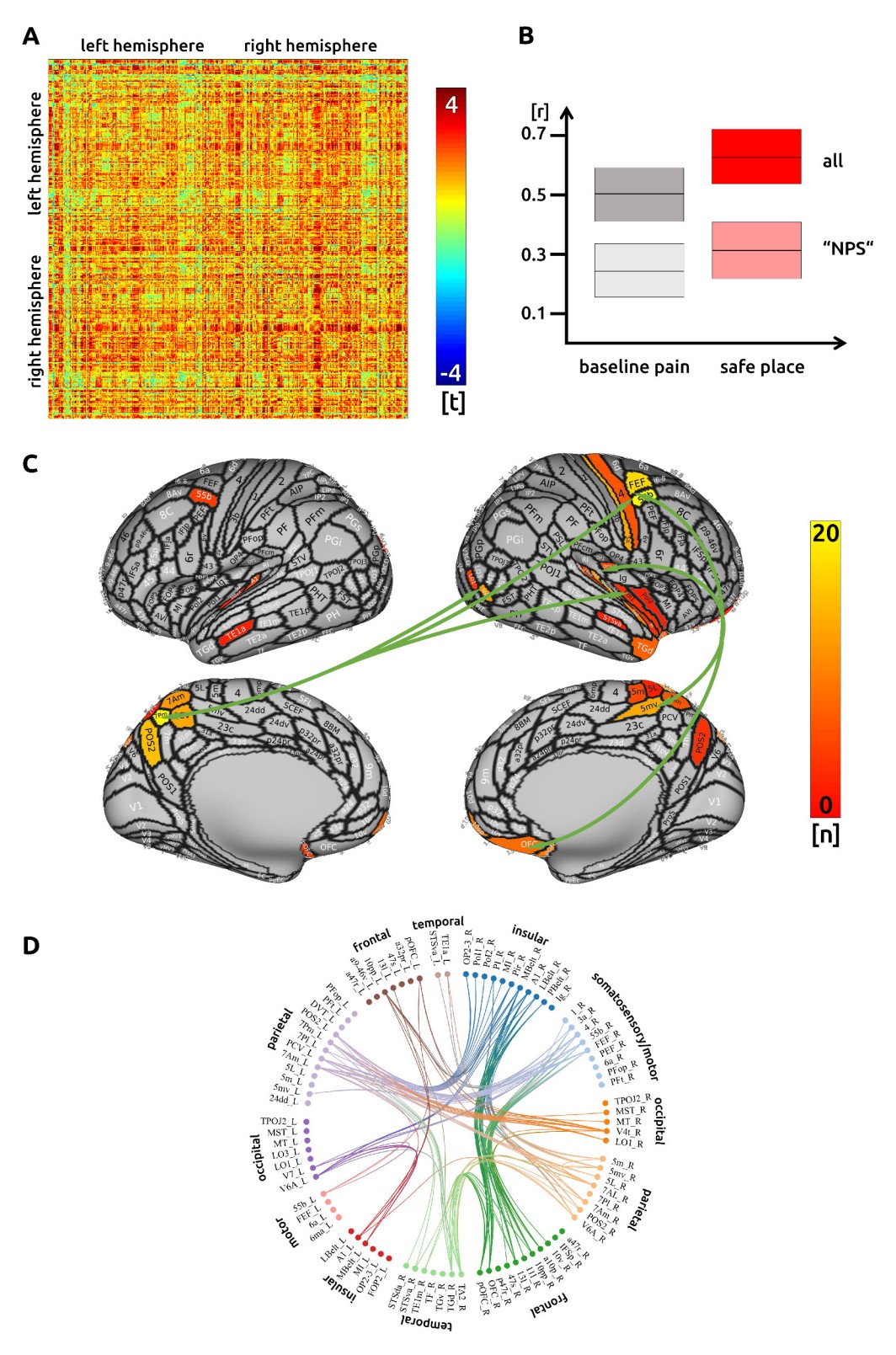

**Figure 2.** Safe place: (A) the confusion matrix shows the statistical results. Each line represents one of the 371 ROIs. The values are mirrored along the principal diagonal of the matrix. A single red dot represents the varying connectivity between two specific brain regions and indicates that a stronger cortical connectivity in a single trial is related to a decrease in pain perception (performance encoding). These findings are the result of the higher connectivity in the trials of the imagination task compared to unmodulated pain trials. (B) Data from the confusion matrix averaged across all subjects,

*Figure 2 continued on next page*

*Figure 2 continued*

connections and trials (mean ± standard deviation; for illustration purposes only). The darker boxes show the average across all connections, the lighter boxes represent the averaged connections within the "NPS". (C) Depiction of the cortical regions as defined by the Glasser parcellation; the arrows show the best connected regions; the left parietal cortex and right premotor areas have the most connections where connectivity changes are shown to significantly modulate pain intensity. Only regions with at least three significant connections (n > 2) are included in the cortical map. (D) For the circular plot, created with Brain Data Viewer (link) we selected 89 regions that showed at least three significant connections in any of the three conditions. For more detailed information on the exact connections see the source data files.

## Discussion

Here, we aimed to explore how functional and structural connections in the brain contribute to executing cognitive tasks that attenuate pain (*Devine and Spanos, 1990*; *Schulz et al., 2019*) by utilising a single-trial analysis approach afforded by ultra-high field imaging. Across three experimental conditions, 20 healthy participants were asked to (a) count backwards, (b) imagine a safe and happy place, and (c) apply a cognitive reappraisal strategy, whilst receiving reasonably long-lasting cold pain stimuli. All strategies resulted in significant pain relief when compared to the unmodulated pain condition. The participants were assumed to execute the tasks with considerable motivation as their effort would be rewarded with decreased pain intensity and pain unpleasantness (*Huskey et al., 2018*; *Pochon et al., 2002*). We applied a whole-brain approach on the basis of brain parcellation definitions (*Glasser et al., 2016*) and explored connectivity patterns during single attempts to attenuate pain. We further explored whether functional connections are facilitated by axonal fibre connections, as measured using DTI fibre tracking.

The functional connections identified here are interpreted in light of our recently reported study that determined the changes in regional BOLD activity during successful single attempts to attenuate pain – drawn from the same data set used for this connectivity analysis (*Schulz et al., 2019*). In that study, non-imaginal reinterpretation (reappraisal) and the imaginal strategy (safe place) predominantly involved modulation of the anterior insula, the non-imaginal distraction task (counting backwards) predominantly modulated the central operculum, whereas tasks that involved distraction from pain (counting, safe place) modulated posterior cingulate cortex activity. Here, we show for the first time at a single-trial level how connectivity between brain regions relates to the effect of task performance on pain attenuation across the entire brain. It is still not completely clear how cognitive tasks are executed through functional interactions between brain regions to modulate painful experiences. Additionally, there is no seed-based whole-brain investigation on how the experience of pain fluctuates across trials driven by the variable strength of cortical connections.

It should be noted that for each of the regions from the Glasser parcellation, we computed a subject-wise principal component analysis (PCA) to relate the time-course to other brain regions. Although this approach - to analyse the components with the highest explained variance - is justified, we note that there are other approaches, for example to run an ICA on concatenated data across all subjects that future studies might explore.

Across all cognitive interventions, our results revealed a global *increase* of connectivity throughout the cerebral cortex for all three interventions: higher functional connectivity was related to particularly effective single attempts to attenuate pain. Therefore, the unmodulated pain trials - which were experienced as considerably more painful - exhibited a lower functional connectivity compared to pain trials during cognitive tasks. This finding has two implications:

First, increased connectivity in *task-related* regions is necessary to effectively execute the respective cognitive tasks. S*econd*, contrary to our hypothesis and previous findings, increased connectivity with brain regions commonly associated with generating pain perception (e.g. insular cortex, ACC, or somatosensory cortices; *Tracey and Mantyh, 2007*) is related to more effective attenuation trials resulting in decreased pain ratings. We suggest that this increased connectivity is required to actively suppress *activity* in regions known to contribute to pain processing, as previously reported for these tasks (*Schulz et al., 2019*). We do not know the nature of such regional suppression; however, there are GABAergic neurons in subregions of the insular cortex (*Watson, 2016*; *Thiaucourt et al., 2018*), and these might possibly contribute. Besides the general observation of increased connectivity, which is linked to higher task performance and lower pain, each cognitive intervention has its own pattern of intra-cortical connectivity.

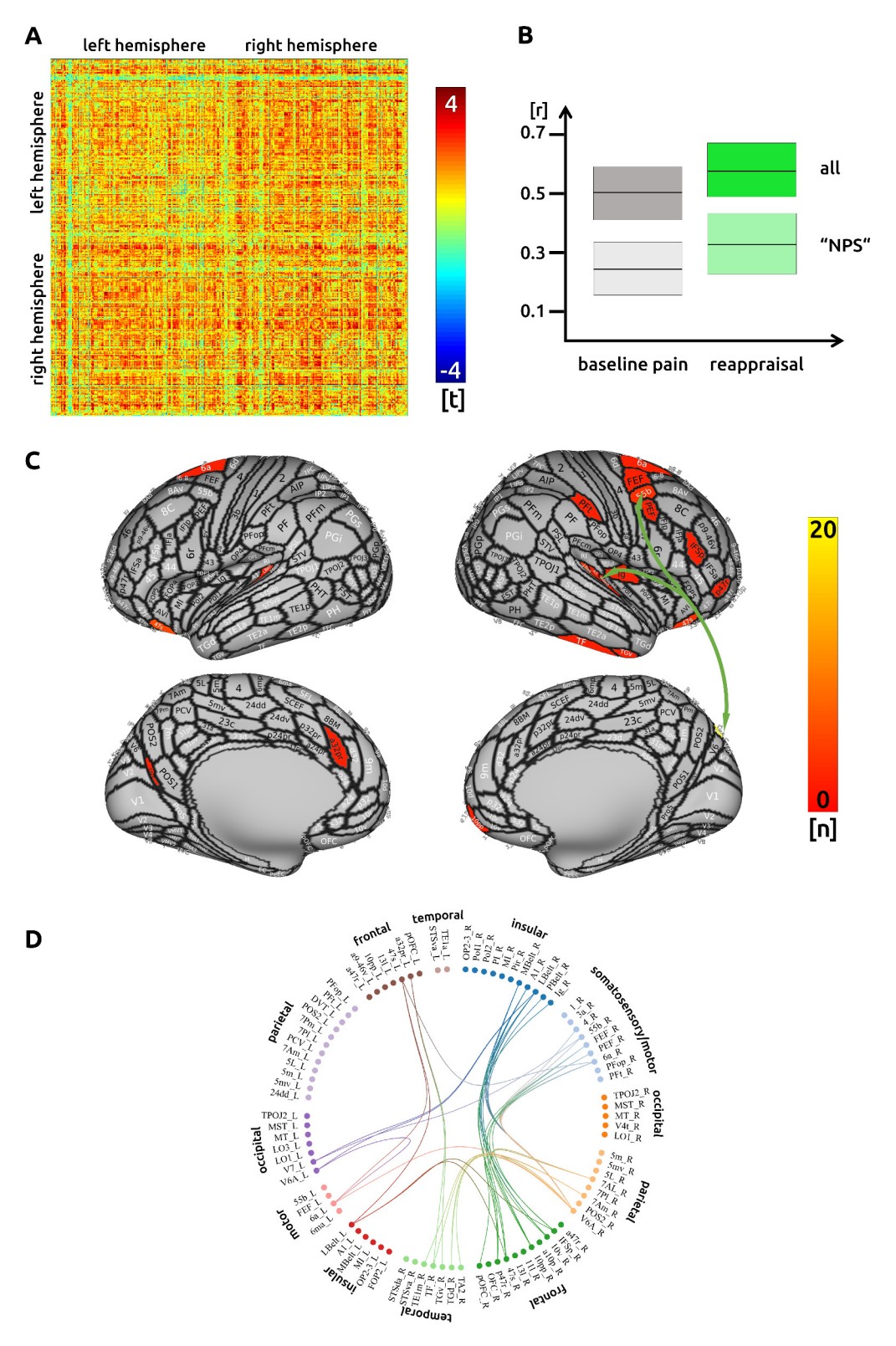

**Figure 3.** Reappraisal: (**A**) the confusion matrix shows the statistical results. Each line represents one of the 371 ROIs. The values are mirrored along the principal diagonal of the matrix. A single red dot represents the varying connectivity between two specific brain regions and indicates that a stronger cortical connectivity in a single trial is related to a decrease in pain perception (performance encoding). These findings are the result of the higher connectivity in the trials of the reappraisal task compared to unmodulated pain trials. (**B**) Data from the confusion matrix averaged across all subjects,

*Figure 3 continued on next page*

*Figure 3 continued*

connections and trials (mean ± standard deviation; for illustration purposes only). The darker boxes show the average across all connections, the lighter boxes represent the averaged connections within the "NPS". (C) Depiction of the cortical regions as defined by the Glasser parcellation; the arrows show the best connected regions; the region V6A in the parieto-occipital cortex has the most connections where connectivity changes are shown to significantly modulate pain intensity. Only regions with at least three significant connections (n > 2) are included in the cortical map. (D) For the circular plot, created with Brain Data Viewer (link) we selected 89 regions that showed at least three significant connections in any of the three conditions. For more detailed information on the exact connections see the source data files.

## Counting

Within the 171 functional connections identified as significantly altered during the counting task, we found a number of very well-connected regions that contribute directly or indirectly to the reduction of pain intensity. These regions are located in the parietal and occipito-temporal cortices, they overlap with regions reported previously as modulated during counting tasks (*Johansen-Berg and Matthews, 2002*; *Schulz et al., 2019*), and yet our results extend these observations. Increased connectivity during counting occurred between regions long associated with processing painful experiences, for example: the middle insular cortex and the primary and the secondary somatosensory cortices. The execution of the counting task is suggested to require visual support by imagining the numbers in space (*Amalric and Dehaene, 2016*). Visual areas in the *left* occipito-temporal cortex connect to and suppress right parietal opercular areas. We also found visual support located in the *right* occipito-temporal cortex that is functionally connected to parietal areas, which in turn suppress the activity in insular sub-regions.

The highest number of connections to other brain regions during the counting task was found for the right middle insular cortex. Although our analyses do not allow for any assumptions on directionality, the functional connectivities between left parietal areas (high BOLD activity) and right insular sub-regions (low BOLD activity) support a likely suppression effect on these insular regions (*Schulz et al., 2019*). The highly connected superior temporal region is suggested to support the retrieval of mathematical knowledge (*Polspoel et al., 2017*).

Disrupted connectivities during the counting task were observed for the right temporo-parieto-occipital junction (TPJ) to the right posterior insula, as well as to temporo-occipital areas. Given the involvement of the TPJ in attentional processing (*Kucyi et al., 2012*; *Mars et al., 2012*), elevated focus on the task may have decreased nociceptive transmission to the posterior insula during task execution but increased the transmission for unmodulated pain trials (*Schulz et al., 2019*). We did not find any significantly altered connectivity with subcortical regions that have been linked to pain modulation, such as the thalamus, the amygdala, nucleus accumbens or the PAG.

## Safe place

Out of 210 significant connections, we found regions in the left and right parieto-occipital cortex to be particularly connected to other brain regions. There are four major hub regions that contribute to the imaginary task through strong functional connections. Notably, the bilateral parietal cortex (area five right, area seven left) is functionally connected without a concomitant rise of regional BOLD activity (*Schulz et al., 2019*). This finding highlights the complementary roles that both regional activity changes or connectivity changes between regions play when modulating painful experiences: active exchange of information with potentially low-scale modulations of cortical activity that are not causing large and measurable metabolic effects. Possibly, the strong connectivity pattern between left parietal and right insular sub-regions suggests an active suppression of these insular regions initiated by the parietal cortex, as reflected by increased functional connectivity between these regions.

We found well-connected regions in the precentral gyrus: area 55b has been shown to be active during listening to stories in the language task of the Human Connectome Project dataset (*Glasser et al., 2016*). Therefore, the increased connectivity in area 55b may be related to the narrative aspects of the imaginary task, in which the participants may recall being actively involved in an event of pleasure and happiness. The premotor and motor areas in the precentral gyrus in particular may reflect the motor aspect of the imagination task (*Szameitat et al., 2007*; *Xie et al., 2015*). They are connected to orbitofrontal regions that have been highlighted as important in mediating predictive cues about pain in conditioning and placebo studies (with involvement of the descending pain modulatory inhibitory network; *Atlas et al., 2010*; *Bingel et al., 2007*; *Eippert et al., 2009*) as well

as studies highlighting the role of the orbitofrontal cortex in threat valuation of pain during attentional guiding in clinical studies (*Scharmüller et al., 2014*). The strong connectivity of the auditory 'belt' regions is interesting and possibly suggests its important contribution to the imaginary task; activity here might reflect not only current sensory input (*Moerel et al., 2014*) but also previous sensory experience (*Meyer et al., 2010*).

For the safe place condition only, we found that subjects' ability to functionally utilise certain pathways is mediated by axonal fibre connections, as measured using DTI tractography. These anatomical characteristics are suggested to help the participants better attenuate pain. This applies especially to connections between middle frontal regions (IFSp and 8C) and the secondary somatosensory cortex (SII). Further functional connections that are facilitated by axonal fibre connections project from the frontal cortex to memory-related limbic areas (presubiculum of the hippocampus, entorhinal cortex); these might facilitate memory retrieval for the imagination of pleasant and complex scenes (*Braskie et al., 2009*; *Dalton and Maguire, 2017*; *Hodgetts et al., 2017*; *Montchal et al., 2019*).

## Reappraisal

From the 70 significant functional connections found during cognitive reappraisal, the most highly connected region is located in the higher order visual cortex, area V6A. This region is mainly interconnected with several insular and frontal premotor areas. Area V6A is known to contribute to spatial object localisation; a study on monkeys shows that V6A cells are active when executing reaching movements independent of visual or oculomotor processing (*Fattori et al., 2005*). These cells have also been found to encode body-centred spatial localisation (*Hadjidimitrakis et al., 2014*). The use of V6A and its connection to other brain areas could help the participants - as required by the task - to focus on the stimulated body site. However, this focussing should be considered as a prerequisite and does not necessarily imply any pain attenuation. In fact, focusing on pain generally increases pain perception and pain-related cortical activity (*Hauck et al., 2007*; *Peyron et al., 1999*) – and distraction or mind-wandering away from pain generally engages descending modulatory systems to alleviate pain (*Kucyi et al., 2013*; *Tracey et al., 2002*). It is not yet clear how reappraisal modulates painful experiences, although a recent study using an imaginary reappraisal task confirms a lack of engagement of descending inhibitory opioidergic pathways (*Berna et al., 2018*). Neuroimaging studies exploring how mindfulness modulates painful experiences are growing and might give us additional insight (*Zeidan et al., 2019*).

Along with V6A, parts of the right prefrontal Brodmann area 10 and the bilateral orbitofrontal and the bilateral Brodmann area 47 account for more than half of the significant connections during cognitive reappraisal. These frontal regions have been linked to a number of cognitive processes (*Snow, 2016*), with BA 10 playing a role in emotion regulation (*Golkar et al., 2012*; *Hallam et al., 2015*) and orbitofrontal BA47 in decision making (*Padoa-Schioppa and Conen, 2017*).

The reappraisal condition seems to be more complex as it has the most regions (30 out of 70), compared to counting and 'safe place', with only one significant connection. Therefore, as found in the present investigation, the connections from the inferior frontal cortex, the anterior cingulate cortex, the frontal pole, and orbitofrontal cortex are all harnessed during cognitive reappraisal (*Buhle et al., 2014*; *Tracey, 2010*) in order to ultimately attenuate the experience of pain (*Schulz et al., 2019*).

## Analysing functional connections between brain regions involved in pain processing

We found almost exclusively a lower functional connectivity for trials and conditions of higher pain intensity. There are important design and methodological differences to previous studies that likely explain why we unmasked these observations. In neuroimaging, functional connectivity is considered a joint phase-locked oscillation of spatially distinct cortical regions. Task-based connectivity analyses predominantly utilise a seed-based approach to determine the functional connectivity between a predefined seed region and one or more distant brain regions; such analyses can only take into account the short period during which a task is being executed. However, exact connectivity measures between brain regions would require a sufficient number of samples to quantify the joint in-phase increases and decreases of the BOLD response. In order to estimate a reliable measure of

connectivity, we applied a relatively long time window (~30 s) for inflicting pain, for executing the cognitive task, and for reliably determining the connectivity of a single trial. The extended stimulation and the restriction of the analysis to the plateau phase resulted in a solid data basis (330 time points for each subject for each condition). For comparison, a study by *Villemure and Bushnell, 2009* sampled every 4 s, but analysed a relatively short time window of 5 s painful stimulation in order to investigate connectivity. Another study analysed just a single data point per trial (3 s analysis window, sampling of 3 s) before nociceptive laser stimulation to predict pain intensity (*Ploner et al., 2010*). A further study used three data points per trial for connectivity analyses of an experiment in which the pain stimulations lasted 10 s (*Sprenger et al., 2015*). A repeated stimulation at the frequency of the recording (application of 5 s brief laser pain stimuli every 3 s sampled with a TR of 3 s) makes it difficult to separate the connectivity aspects from the phasic increases in BOLD response to repeated stimulation (*Ploner et al., 2011*).

In addition, we used a particular artefact cleaning method recommended for connectivity analyses where there might be task-related movement due to small muscle twitches during painful stimulation. Our focus on the analysis of the stimulation plateau makes the connectivity estimates unaffected by the statistical design matrix. Therefore, the different methodological approaches have likely contributed to our finding something different to other studies published to date – whereby high levels of pain increase cortical connectivity between pain processing brain regions (*Sprenger et al., 2015*). Villemure and Bushnell and Ploner and colleagues found a stronger connectivity in pain processing brain regions for conditions that increased the intensity of pain (i.e. increased attention, more negative emotion). The connectivity of the inferior frontal cortex for an emotional condition and the connectivity of the superior parietal cortex and the entorhinal cortex for the attentional condition were found to modulate pain (*Ploner et al., 2011*; *Villemure and Bushnell, 2009*). Our results for increased connectivity in pain processing brain regions during decreased pain experiences were corroborated by the additional analysis that focussed on connectivity changes within brain regions related to the "NPS" (*Wager et al., 2013*). For all three cognitive tasks, we confirmed a significantly increased connectivity between "NPS" brain regions during successful pain attenuation.

Other studies investigated connectivity in the descending pain modulatory system and observed an increase of connectivity between the rostral ACC and the PAG during a pain-relieving placebo intervention (*Eippert et al., 2009*). Given the challenges of imaging mid-brain regions at seven tesla when acquisition is optimised for whole brain data acquisition, this finding could not be replicated in any of the present conditions – also due to the strict correction for multiple comparisons (*Winkler et al., 2014*). For exploratory purposes, however, when the statistical threshold was lowered, we do find an increase of functional connectivity for all three conditions from the PAG to regions that contribute to pain processing, such as the anterior ventral insula, the midcingulate cortex, the posterior ACC, and the nucleus accumbens (all $p < 0.05$, uncorrected).

In addition to studies examining the cognitive or contextual modulation of pain, other work has investigated connectivity changes in response to different applied intensities of pain stimulation. *Sprenger et al., 2015* found an increase of connectivity in subcortical nuclei for the higher of two pain conditions. Similarly, an increased connectivity has been found in response to cold pain stimulation (*Wilcox et al., 2015*). The authors reported a significant correlation across the entire time course of the experiment between predefined regions that are known to be involved in the processing of pain. As discussed above, our data show that a decrease in pain is predominantly related to an increase of cortical connectivity in *both* brain regions involved in pain processing and task-related brain regions (e.g. subregions in the frontal and parietal cortex).

## Summary

Using various cognitive strategies, we investigated what cortical connections contribute to those trials effective in attenuating pain. In contrast to previous research, we revealed an increased connectivity for single attempts that resulted in decreased pain. This applies to both classical pain processing regions (e.g. insula, cingulate cortex, and somatosensory cortices) and task-related brain regions. Although we found different connectivity patterns for each of the interventions, the overall increased connectivity was common to all three. Going forward, the present findings might help our understanding and delivery of cognitive interventions for the alleviation of pain in the clinical setting, by focussing on the relevance of connectivity patterns.

## Materials and methods

Twenty two healthy human subjects (18 female/4 male) with a mean age of 27 ± 5 years (21–37 years) participated in the experiment. Two of the female subjects were excluded as a result of insufficient data quality. All subjects gave written informed consent. The study was approved by the Medical Sciences Interdivisional Research Ethics Committee of the University of Oxford and conducted in conformity with the Declaration of Helsinki.

The experiment, the participants, and the behavioural data have been described in detail in our previous publication (*Schulz et al., 2019*). The task consisted of four conditions (see *Table 1*) across four separate blocks, where each block comprised 12 trials from the same condition. The first condition was always the unmodulated condition; the order of the conditions with cognitive interventions were randomised using Matlab (randperm). In all conditions and trials the subjects received cold pain stimuli on the dorsum of their left hand delivered by a thermode (Pathway II; Medoc Ltd, Ramat Yishai, Israel). The subjects were prompted to rate pain intensity and pain unpleasantness. A numerical and a visual analogue scale (VAS), ranged between 0 and 100 in steps of five points, was used to assess the pain ratings. The endpoints of the scale were determined as no pain (0) and the maximum pain the subjects were willing to tolerate (100). Single-trial ratings were recorded after each trial.

The thermode temperature for painful stimulation for each subject was determined in an extensive practise session 1 week prior to scanning and was individually adapted to a VAS score of 50. The 40 s of painful stimulation were then preceded by a rest period of 10 s at 38°C thermode temperature. The first 10 s were not included in the analysis. The mean temperature of cold pain application across subjects was 7°C with a standard deviation of 3.6°C. In order to avoid habituation effects, the thermode temperature during painful stimulation was oscillating with ⅛ Hz at ±3°C (*Lautenbacher et al., 1995*; *Stankewitz et al., 2013*).

### Data acquisition

Imaging data were acquired on a 7T Siemens MRI scanner. Each volume comprised 34 axial slices of 2 mm thickness and 2 × 2 mm in-plane resolution with 1 mm gap between slices. The repetition time (TR) was 1.96 s, the echo time (TE) was 25 ms (flip angle 90°), the field of view (FOV) was 220 × 220 mm, and the matrix size was 110 × 110 pixels. A T1-weighted structural image (isotropic 1 mm³ voxel) was acquired for the registration of the functional images to the MNI (Montreal Neurological Institute) template. Two sequences of diffusion tensor images (DTI) were recorded with L >> R and R >> L phase encoding direction. 64 directions were recorded with a TR of 9.3 s, a TE of 63 ms, and an acceleration factor of 2. The length of the edge of the isotropic voxels was 1.2 mm.

### Image processing – pre-processing of functional connectivity data

The data were pre-processed with FSL (*Jenkinson et al., 2012*). The pre-processing of the *functional data* consisted of brain extraction, high-pass filtering with a frequency cutoff of 1/90 Hz, a spatial normalisation to the MNI template, a correction for head motion during scanning registered to the MNI template, and a spatial smoothing (6 mm FWHM). The data were further semi-automatically cleaned of artefacts with independent component analysis (ICA) (*Griffanti et al., 2014*; *Salimi-Khorshidi et al., 2014*). The number of components had been set a priori to 200. Artefact-related components were removed from the data. The design matrix for painful stimulation, including the temporal derivative, were then regressed out from the data in Matlab (The Mathworks, USA).

**Table 1.** Conditions and Instructions.

| (0) Pain, non-modulated | Concentrate only on the pain. |
| --- | --- |
| (A) Attentional shift | Count backwards from 1000 by sevens. |
| (B) Imaginal strategy | Imagine that you are in a safe and happy place that you know very well. That place has the colours you like and you hear the music you like. There are only people around that you want to have around you. You feel well and comfortable. |
| (C) Cognitive reappraisal | Concentrate on the cool and tingling sensations in your arm and reinterpret these sensations as not painful. |

## Image processing – pre-processing of structural connectivity data

Pre-processing of DTI data was performed using FSL. FSL pre-processing included (i) correcting susceptibility induced distortions ('topup'), (ii) skull stripping ('bet'), and (iii) corrections for eddy currents and head motion ('eddy'). We finally (iv) determined the strength of structural connectivity between cortical regions ('bedpostx' and 'probtrackx') defined by the Glasser atlas. For tractography, we used the 'one-way condition', with 5000 samples, 2000 steps per sample, a step length of 0.5, and a fibre threshold of 0.1. The number of streamlines quantifies the strength of structural connectivity for two brain regions of a subject.

## Image processing - extraction of regions of interest data

The time series of functional volumes were converted to MNI space and subsequently projected to surface space by using the 'Connectome Workbench' package. We used a template that allowed to project from 3D standard MNI space to 2D surface space. Regions of interest (ROIs) were defined by subdividing the cortical surface into 180 regions per hemisphere (*Glasser et al., 2016*). Six further regions (five bilateral) that are important for the processing of pain, such as the PAG, the thalamus and the amygdala, were also included. Latter ROIs were based on the Oxford Atlas, implemented in FSL.

## Image processing - computation of single trial functional connectivity scores

The time courses for all voxels of cortical activity for a specific region of the Glasser Atlas, for example the middle insula, were extracted. We computed principal component analyses (PCA) separately for each ROI and subject and selected the first component (Matlab, The MathWorks, Inc, USA). The plateau phase of the last ~30 s of painful stimulation (15 data points) has been extracted from each region and trial for each subject and condition. Outliers were removed from the data by using the Grubbs' test (*Grubbs, 1950*). These 15 data points determined the connectivity for a brain region for a given trial. Correlation coefficients were computed for each trial and for each of the 371 ROIs with the remaining 370 ROIs. The single-trial correlation coefficients were Fisher Z-transformed and fed into group-level statistical analysis.

## Image processing - structural connectivity data

DTI data were also analysed in FSL. The processing steps included a median filter, a correction for susceptibility distortions, and fibre tracking from the same aforementioned brain regions (Glasser parcellation - see above).

## Statistical modelling

The statistical analysis for the connectivity between cortical regions has been performed in Matlab. To explore the relationship of fluctuating cortical connectivity and the variable pain experience, we computed linear mixed effects models (LMEs) that related the single-trial correlation coefficients between two brain regions to the pain intensity scores (*Michail et al., 2016*; *Schulz et al., 2015*). Each condition in the model included the data for the respective intervention plus the trials of the unmodulated pain condition. Using the Wilkinson notation (*Wilkinson and Rogers, 1973*), the model can be expressed as:

$$painrating \sim func\_conn + (1|subject)$$

The included fixed effects (*func_conn*) essentially describe the magnitudes of the population common intercept and the population common slope for the relationship between cortical connectivity and pain perception. The included random effects (1| subject) are used to model the specific intercept differences for each subject.

This single trial approach has two major advantages: *First*, it takes the within-subjects variable performance of the pain attenuation attempts into account; for example a more successful attempt to attenuate pain (as reflected by lower pain ratings) is considered to cause a stronger cortical effect (compared to the unmodulated pain condition) than a less successful attempt. *Second*, it also takes into account the more natural fluctuation of the unmodulated pain trials (without cognitive intervention).

The assessment of the direct difference between the three conditions required an additional model. We compared each experimental condition with the remaining conditions in order to explore areas that are uniquely activated for this condition:

$$painrating \sim func\_conn + func\_conn * condition + (1|subject : condition)$$

The second model introduces an interaction term (fixed effect: *func_conn*\*condition), which represents how the slope of each condition is affected by the different conditions. This interaction term allows us to contrast the different conditions. An individual intercept for each subject and condition was included (random effect: 1|subject:condition).

We further analysed whether individual differences in functional connectivity could be explained by individual structural characteristics of the brain. In other words, we analysed whether the functional connectivity that leads to a single subject's successful pain attenuation is facilitated by that subject's high number of fibre tracts. In a similar vein, a poor functional connectivity that is not able to contribute to pain attenuation might be caused by a low number of fibre tracts.

$$painrating \sim func\_conn : struc\_conn + (1|subject)$$

We considered only functional connections with a t-value >2 as potentially modulated by structural connections.

To correct for multiple comparisons, we applied a randomisation approach. Behavioural data were shuffled and the entire analysis was repeated 5000 times. The highest absolute t-values of each repetition across the whole confusion matrix was extracted. This procedure resulted in right-skewed distribution for each condition. Based on these distributions, the statistical thresholds were determined using the 'palm_datapval' function implemented in PALM (*Winkler et al., 2014*).

## Additional information

### Funding

| Funder | Grant reference number | Author |
| --- | --- | --- |
| Deutsche Forschungsgemeinschaft | 2879/1-1 | Enrico Schulz |
| Wellcome | 090955/Z/09/Z | Irene Tracey |
| Wellcome | 083259/Z/07/Z | Irene Tracey |
| Medical Research Council | G0700399 | Irene Tracey |

The funders had no role in study design, data collection and interpretation, or the decision to submit the work for publication.

### Author contributions

Enrico Schulz, Conceptualization, Resources, Data curation, Software, Formal analysis, Supervision, Funding acquisition, Validation, Investigation, Visualization, Methodology, Writing - original draft, Project administration, Writing - review and editing; Anne Stankewitz, Conceptualization, Writing - original draft, Project administration, Writing - review and editing; Anderson M Winkler, Software, Methodology, Writing - review and editing; Stephanie Irving, Writing - original draft, Writing - review and editing; Viktor Witkovský, Software, Formal analysis, Methodology; Irene Tracey, Conceptualisation, Resources, Supervision, Visualisation, Funding acquisition, Writing - review and editing

### Author ORCIDs

Enrico Schulz (iD) https://orcid.org/0000-0001-8188-380X

### Ethics

Human subjects: Informed consent and consent to publish was obtained in accordance with ethical standards set out by the Declaration of Helsinki (1964) and with procedures approved by the

Medical Sciences Interdivisional Research Ethics Committee of the University of Oxford (REC ref: MSD-IDREC- C1-2014-157).

## Decision letter and Author response
Decision letter https://doi.org/10.7554/eLife.55028.sa1
Author response https://doi.org/10.7554/eLife.55028.sa2

## Additional files

### Supplementary files
- Source data 1. The file contains the results that generated *Figure 1*.
- Source data 2. The file contains the results that generated *Figure 2*.
- Source data 3. The file contains the results that generated *Figure 3*.
- Source data 4. The file contains the results for the analysis on the facilitation of functional connectivity through structural connectivity.
- Source data 5. The file contains the results for the contrast between pain attenuation during counting and pain attenuating during 'safe place'.
- Source data 6. The file contains the results for the contrast between pain attenuation during counting and pain attenuating during reappraisal.
- Source data 7. The file contains the results for the contrast between pain attenuation during 'safe place' and pain attenuating during reappraisal.
- Transparent reporting form

### Data availability
The dataset is available at the Open Science Framework (https://osf.io/tbc2u/). The source data files to generate the figures are included in the submission (Source data 1–7).

The following dataset was generated:

| Author(s) | Year | Dataset title | Dataset URL | Database and Identifier |
|---|---|---|---|---|
| Schulz E | 2020 | Pain Attenuation | https://osf.io/tbc2u/ | Open Science Framework, tbc2u |

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
