## [Decision Letter]

**Acceptance summary:**

This article reports the functional connectivity of brain regions recruited during cognitive strategies employed to attenuate pain. Using a wide range of cutting-edge methodologies, the study helps to understand how brain circuits can be harnessed to control pain. These results are timely and have clinical implications.

**Decision letter after peer review:**

Thank you for submitting your article "Ultra-high field imaging reveals increased whole brain connectivity underpins cognitive strategies that attenuate pain" for consideration by *eLife*. Your article has been reviewed by two peer reviewers, one of whom is a member of our Board of Reviewing Editors, and the evaluation has been overseen by Christian Büchel as the Senior Editor. The reviewers have opted to remain anonymous.

The reviewers have discussed the reviews with one another and the Reviewing Editor has drafted this decision to help you prepare a revised submission.

Summary:

This article reports online changes in functional connectivity based on a whole brain approach in healthy subjects who repeatedly rated pain intensity and unpleasantness in a tonic cold stimulation task while they used cognitive strategies to attenuate pain. This represents a timely and important area in pain research, with strong clinical implications. Methodological assets include three different types of cognitive-attentional tasks, high field imaging at 7T, use of pain levels and connectivity analyses in the plateau phase, counteracting of habituation effects and trial by trial analyses over an extended time period. The scientific findings are interesting and will advance the field.

Essential revisions:

1) Cite relevant work from other researchers in the Introduction, e.g. Davis and colleagues.

2) Summarize relevant previous results from the cited 2019 publication on the same data set.

3) Acknowledge clearly that motivational processes are also involved here and use the term 'pain processing' with care.

4) The summary is excessively geared towards relevance for chronic pain, but since networks differ from those of acute pain, use a caveat regarding the direct transfer of these findings.

5) Explain better what is fundamentally new in the study, since increased connectivity across the brain during cognitive and attentional tasks is per se not surprising.

6) Given the findings on pain attenuation, address thoroughly participation of areas that have been implicated in inhibition of pain in the description of results and Discussion section.

7) Analyze and discuss more thoroughly the identity of cortical regions that undergo inhibition or excitation during the tasks and discuss more concretely the flow of information between cortical and sub-cortical pathways and the strength of their connectivity, which may explain the observed phenotypic effects on pain attenuation.

8) No additional experiments are needed but the authors needs to address especially the specificity issue.

---

## [Author Response]

Essential revisions:1) Cite relevant work from other researchers in the Introduction, e.g. Davis and colleagues.

It is always a challenge to include all relevant references despite a willingness. We have attempted to amend this in the Introduction (e.g. Liang et al., 2012; Misra and Coombes, 2014). The reference list now contains over 50 citations. Consistent with our approach and hypothesis, we based the Introduction on publications that investigate task-based connectivity changes on pain. Due to profound methodological differences, this therefore excludes studies on pain-related functional connectivity that are not task based, i.e. ICA based resting state studies (e.g. PMID: 29994989). It is not possible to compare our findings on fluctuations of the connectivity time course to cortical maps that show the strengths of ICA components. In the present study, we are focussed on within-subject task-based variations, which are not readily comparable to patients’ pain-related ICA components. Other excellent studies, i.e. by the Davis group (PMID: 17314240) investigate how pain affects cognitive processes, which is different again and the opposite of what we have done here. We have made this clearer in the Introduction section by mentioning that ICA-based studies pursue a fundamentally different methodological approach than seed-based connectivity studies on pain and, as such, provide a different physiological window. However, we have now included further relevant literature that use the latter approach.

2) Summarize relevant previous results from the cited 2019 publication on the same data set.

We agree that it might be helpful for the reader to have a short summary of the BOLD findings from this previous publication. We added this to the beginning of the Discussion section.

3) Acknowledge clearly that motivational processes are also involved here and use the term 'pain processing' with care.

We thank the reviewers for raising this key issue. We acknowledge that motivational processes play an important role in the study, whereby an increased cognitive effort might be rewarded with lower pain intensities. We have now added this important point to the beginning of the Discussion section, and thank the reviewer for the excellent suggestion.

4) The summary is excessively geared towards relevance for chronic pain, but since networks differ from those of acute pain, use a caveat regarding the direct transfer of these findings.

Our motivation for this gearing was to stimulate further and similar work in pain patients. Also, the study was inspired by our desire to better understand clinical cognitive interventions on chronic pain. But we agree this could be dialled down and so we have attempted to do so.

5) Explain better what is fundamentally new in the study, since increased connectivity across the brain during cognitive and attentional tasks is per se not surprising.

We thank the reviewer for the comment and would like to take the opportunity to make clearer what is fundamentally new and to emphasise the novelty of our findings. To our knowledge, there is no study on pain nor on applied cognitive interventions that uses a whole-brain seed-based functional connectivity approach. We pursued the *whole brain* approach on cortical connectivity to ensure we identified all effects, as well as to rule out any suspicion of HARKing. As this is the first study examining how cognitive interventions are related to connectivity profiles in the context of pain, there was no prior work to guide us. As a consequence, our hypotheses were limited to the following:

↑ pain ~ ↑connectivity (hypothesised for “pain” regions but not confirmed)

↓ pain ~ ↑connectivity (hypothesised for “task” regions; confirmed).

We have made clearer in the revised text the novelty of our findings. As hypothesised, we found an increased functional connectivity for regions that can be associated with the execution of the cognitive tasks. However, in contrast to our hypothesis we found a higher functional connectivity during the high-pain condition for regions that are known to be associated with pain. This latter aspect, we have now expanded to include an additional analysis of brain regions within the neurological pain signature (NPS) network (see below). The results of which confirm and support our original findings and strengthen our confidence in the unexpected and novel observation of increased connectivity leading to decreased pain.

6) Given the findings on pain attenuation, address thoroughly participation of areas that have been implicated in inhibition of pain in the description of results and Discussion section.7) Analyze and discuss more thoroughly the identity of cortical regions that undergo inhibition or excitation during the tasks.

We thank the reviewers for encouraging us to expand on this point in our revised text. We assume the reviewer is explicitly asking about areas involved in the descending pain modulatory system, as well as other brain regions known to play a role in driving pain attenuation; and to then further discuss regions inhibited as a consequence? The challenge we faced was that out of 68635 possible connections, 171 connections were significant for counting, 210 for "safe place", and 70 for reappraisal. In an attempt to distil the complexity of these results, and graphically illustrate them for a more general audience, we have revised our text and figures. We have clarified also that the relatively large number of significant connections requires us to focus on the main hubs with the highest number of significant connections, in addition to discussing known or expected brain regions implicated in pain inhibition.

To address the reviewer’s request, we now discuss activity findings from the previous publication (that centred on regional BOLD activity increases and decreases in brain regions that are involved in pain inhibition – see Schulz et al., 2019) in the light of these results on connectivity changes. Further, we have now explicitly analysed the connectivity only within brain regions as part of the neurological pain signature (NPS; Wager et al., 2013). For this analysis we averaged 40 bilateral insular, opercular, and cingulate regions, the bilateral thalamus plus the PAG. We averaged the connectivity across these 81 regions for each single trial and confirmed that connectivity increases between these regions when pain trials are successfully modulated to attenuate pain compared to unmodulated trials (counting: t=2.40 (p<0.05); safe place: t=3.35 (p<0.001); reappraisal: t=3.19 (p<0.005)). This additional data is now included and the figures modified accordingly. We hope these changes and additions adequately address the helpful comment.

Discuss more concretely the flow of information between cortical and sub-cortical pathways and the strength of their connectivity, which may explain the observed phenotypic effects on pain attenuation.

Thank you for asking us to discuss this more fully, as it’s clearly an important aspect. We can confirm that there was no supra-threshold connectivity effect to subcortical areas, but we did find connections slightly below the statistical threshold for all three conditions. We have made that clearer in the manuscript. Due to the (likely expected) subthreshold result, we believe cortical-subcortical pathways should be assessed in a subsequent, preregistered study that has clear predefined hypotheses. In order to support the development of such hypotheses for follow-up studies, we have now discussed sub-threshold connectivities greater than t=2.5 that were found for all three conditions.

8) No additional experiments are needed but the authors needs to address especially the specificity issue.

We are not completely sure what the reviewer is asking; however, we have done our best to address their request. We ran a conjunction analysis in order to determine whether there are connections commonly used by all three conditions. The analysis of specificity is a mirrored concept to investigate whether a process is specific to a task (see rResults section).

We found that there are indeed processes that are unique to one strategy but absent in the two others. We added these results to the manuscript. However, the current study investigates only three out of many pain attenuation strategies. Given the lack of knowledge on the cortical processes of the many other strategies, we would speculate that it is unlikely a process is specific only to the strategies we have used here (i.e. and not found in other cognitive pain attenuation strategies). We have made this point clear in the revised manuscript and hope it meets with the reviewer’s approval.